# Proposed Questions to Assess the Extent of Knowledge in Understanding the Radiology Report Language

**DOI:** 10.3390/ijerph191811808

**Published:** 2022-09-19

**Authors:** Mohammad Alarifi, Abdulrahman M. Jabour, Min Wu, Abdullah Aldosary, Mansour Almanaa, Jake Luo

**Affiliations:** 1Health Informatics & Administration Department, College of Health Sciences, University of Wisconsin, Milwaukee, WI 53211, USA; 2Radiological Sciences Department, College of Applied Medical Sciences, King Saud University, Riyadh 11451, Saudi Arabia; 3Health Informatics Department, Faculty of Public Health and Tropical Medicine, Jazan University, Jazan 45142, Saudi Arabia; 4Body Imaging Department, Medical Imaging Administration, King Fahad Medical City, Riyadh 12231, Saudi Arabia

**Keywords:** informatics, imaging informatics, patient awareness, radiology report

## Abstract

Radiotherapy and diagnostic imaging play a significant role in medical care. The amount of patient participation and communication can be increased by helping patients understand radiology reports. There is insufficient information on how to measure a patient’s knowledge of a written radiology report. The goal of this study is to design a tool that will measure patient literacy of radiology reports. A radiological literacy tool was created and evaluated as part of the project. There were two groups of patients: control and intervention. A sample radiological report was provided to each group for reading. After reading the report, the groups were quizzed to see how well they understood the report. The participants answered the questions and the correlation between the understanding of the radiology report and the radiology report literacy questions was calculated. The correlations between radiology report literacy questions and radiology report understanding for the intervention and control groups were 0.522, *p* < 0.001, and 0.536, *p* < 0.001, respectively. Our radiology literacy tool demonstrated a good ability to measure the awareness of radiology report understanding (area under the receiver operator curve in control group (95% CI: 0.77 (0.71–0.81)) and intervention group (95% CI: 0.79 (0.74–0.84))). We successfully designed a tool that can measure the radiology literacy of patients. This tool is one of the first to measure the level of patient knowledge in the field of radiology understanding.

## 1. Key Points

The level of radiology literacy in patients is important.Patient awareness of radiology contributes to an increase in treatment efficiency.A tool that measures radiology literacy in patients is needed.

## 2. Introduction

Patients are becoming the end-readers of the radiology reports with their referring physicians [1,2]. Patient access to radiology tests empowers the patient and allows the patient to be more informed and engaged in his or her healthcare [3,4,5]. The benefits of patient portals that allow patients to access their medical information, including radiology tests and results, have been well studied [3,6,7,8,9,10]. Although many patients might be able to access their reports, patients are often unable to understand the radiology reports. This lack of understanding limits patient engagement in treatment or self-management. Ultimately, these limitations restrict the overall control that a patient has over his or her own health. While improving patients’ level of radiology reports understanding is desired, any intervention should be proceeded by measuring the level of radiology literacy to enable the objective assessment of its impact.

This situation shows that there is a need to develop a tool that can assess patients’ radiology literacy. The extent to which patients are able to read and understand the radiology reports is a reflection of their general radiology literacy [11]. Studies have shown that there is a close relationship between identifying public health awareness and ongoing research [12,13]. There is interest in and a need for developing health literacy tools for conditions such as diabetes and high blood pressure [14,15,16]. A study designed a 10-question tool to measure awareness of blood pressure [14] and another study used the same tool to measure hypertension awareness in individuals living in rural areas of China [17]. This study produced important results that would guide the government in developing healthcare approaches for those who have hypertension in these geographic areas. Likewise, many studies have shown that patient participation in the diagnosis can help to improve medical journey efficiency [18,19,20,21]. Studies have found that one of the most important obstacles in patient involvement is the extent of patient awareness [20,21]. Low patient radiology awareness can cause discomfort and anxiety.

We believe that there is a gap in the literature regarding tools that measure patients’ radiology awareness. There has been an increasing number of studies designed to explore the extent to which radiologists want patients to participate in the treatment process. Studies focusing on patients in this field have previously looked at patient desires and needs without measuring patient knowledge of radiology [22,23,24]. A radiology literacy assessment tool can study patient awareness of radiology. In this study, we designed a Radiology Literacy Tool (13) tool to measure patients’ levels of radiology understanding.

## 3. Methods

### 3.1. Tool Devolopment

To develop the RLT, a set of questions to measure the extent of knowledge of medical radiology in patients is presented in Table 1. The questions were worded carefully to avoid biased, leading questions; assumptions; or loaded questions, which might influence the results. The questions were also reviewed multiple times and tested following the design principles published by Iarossi, G. (2006) [25]. The questions were based on the results of a study we published about patient concerns [26]. The study about patient concerns explored topics that were asked on social media about medical radiology. For validation, we adopted Lawshe techniques for content and face validation [27,28]. A group of radiologists and a radiology specialist were consulted to finalize the latest draft of the questions presented in our radiology knowledge assessment tool. This assessment method was designed to cover various radiology pillars including basic medical terminology, image content, instructions, and radiation protection.

### 3.2. Study Design and Data Collection

This study is part of a larger study in which we sought to evaluate a new design by comparing it with the current radiology report design [29]. We used Amazon’s MTurk platform to distribute the survey. There were 616 participants. Each participant received $0.40, which was paid by the author. The participants were divided into two groups: the intervention group and the control group. The intervention group was given the new radiology report design while the control group was given a radiology report using the current design. The assignments were provided using a random computer distribution. The questionnaire was uploaded onto the Qualtrics platform and linked to the Amazon MTurk platform for distribution. The sample size for each group had to be at least 257 individuals for the alpha to be 0.05 and the desired power to be 0.80. The control group had 320 participants and the intervention group had 296 participants. The study included participants from all 50 states in the United States. The participants needed to be fluent in English to complete the survey. In the consent form, participants were told that their privacy would be protected and that they were free to withdraw at any time. Table 2 shows the participant demographics. The study was approved, and the IRB number is 20.230.

### 3.3. Study Design and Analysis

The total number of participants was 616, with 320 participants in the control group and 296 participants in the intervention group. The unequal number of groups was due to removing the subjects with missing data. Therefore, a chi-squire test was conducted (Table 2) for groups comparison. We also selected non-parametric tests as they are more suited for the unequal group size [30].

The tool was examined by comparing each question with the Reports Comprehension Quiz (RCQ) (Table 3). We used Spearman correlation to measure the correlation between each question in the RLT tool and the overall RCQ test score. The participants were divided by a computer into two groups. Each group was given a different radiology report design (shown in Figure 1, Figure 2, Figure 3, Figure 4 and Figure 5) [29]. After being divided into groups, the same quiz was given to both groups to measure the extent of comprehension of the two reports (Table 3).

To identify the RLT questions indicating patients’ literacy, we assessed the correlation between each of the 13 questions in the RLT (Table 1) and the total score of RCQ questions (Table 3) using Spearman correlation coefficients. Three questions were excluded from the RLT question list (questions 5, 9, and 13) because the *p*-value for the questions was greater than 0.05 (Table 1). Moreover, we used Receiver Operating Characteristics (ROCs) to compute the sensitivity and specificity of the RLT. There were two variables used, the total RCQ score and the total RLT score. The RCQ results were divided into two groups based on the total score: 0–2 and 3–5 (Figure 6). To calculate the total RLT score, we assigned a value of 10 for each question. Respondents could achieve a maximum of 100 points. A similar calculation was used in other studies to measure literacy in various health fields [14,15,16,17]. Statistical analyses were performed using SPSS software.

## 4. Results

Participant characteristics are presented in Table 2. There are no discernible differences between the two groups’ demographics other than gender. The majority of the participants were under 49 years of age. The percentage of males was higher than the percentage of females, where males made up 58.35% of the participant population. The majority of people in both groups had attained a level of education that was a university degree or higher; this information is shown in Table 3. The radiology report was presented in English and all participants were required to both speak and read English fluently. The proportion of participants who had English as a primary language was higher in both the intervention and control groups, at 88.9% and 91.3%, respectively. There were income differences in the participants ranging from those who made more than $10,000 a year to more than $80,000 a year. The health and smoking statuses of the participants were also noted.

Table 1 showed the Spearman coefficient and *p*-value for each question by finding the relationship between each question and the test results. Three questions (questions five, nine, and thirteen) were excluded because the *p*-value was >0.05. Our RLT consisted of 10 questions. When calculating the relationship between the new tool of radiology literacy and the quiz results for the intervention group and control group, we obtained a value of (Spearman 0.522, *p* < 0.001) and (Spearman 0.536, *p* < 0.001), respectively. Figure 6 reveals that the RLT showed a good ability to measure the awareness of radiology (area under the receiver operator curve (95% CI: 0.72 (0.62–0.82)).

Table 4 shows the average score of radiology literacy and the *p*-value for each demographic characteristic. The *p*-value was statistically significant in each demographic characteristic. Moreover, Table 4 shows that radiology literacy increases based on education level. Additionally, radiology literacy was higher for those with income less than $10,000 with an average of 51.97. For those who have an income between $20,000 and $39,000, the rate decreases to 48.54. The radiology literacy rises to 59.82 for people with income between $60,000 and $79,999 and was higher for those who make more than $80,000 at 61.10. Radiology literacy among smokers was lower than in non-smokers at 46.90 and 57.30, respectively. The outcome of those with chronic diseases was to have lower radiology literacy than those who did not have diseases at rates of 50 and 55.08, respectively.

## 5. Discussion

A review of previous studies did not uncover a study that designed a tool that would specifically measure radiology literacy. The closest study was one that studied health literacy in vascular and interventional radiology knowledge of patients [11]. The materials available to vascular patients were collected from two providers, 25 resources from the Cardiovascular and Interventional Radiology Society of Europe (CIRSE) website and 31 resources from the Society of Interventional Radiology (SIR) website. Following the collection of these materials, 65 articles were analyzed for their specific level of readability using the following 10 quantitative scales: Flesch–Kincaid Grade Level, Flesch Reading Ease, Simple Measure of Gobbledygook, New Fog Count, Coleman–Liau index, Gunning Fog Index, Raygor Readability Estimate, Fry Graph, and New Dal–Chall. The study concluded that the reading and scientific knowledge level of patients was often too low for them to fully understand the topic. Our study took the opposite approach and looked at the subject matter of the reports and other materials being provided to patients. This study did not allow us to determine the radiology awareness or knowledge level of a specific population in society, such as those who live in urban or rural settings. Our study looked for studies that set standards for overall health literacy. One of the studies examined was a study that measured hypertension literacy [14]. There were other studies that looked at different medical topics including diabetes literacy. The radiology field had many studies that were concerned with increasing the productivity of radiologists and speeding up the report development process. No study in the radiology field examined radiology literacy in patients.

In this study, we designed a tool to determine the extent of radiology literacy among patients. The questions were developed by a group of specialists in the radiology and health informatics fields. We tested the tool by presenting it to two groups, the intervention and control groups. The same case was presented using different designs. The first design is currently widely used in the medical field and the second design was modified in a way to make the report simpler without changing the content. Radiology literacy was measured using a 10-question tool. Three questions were excluded from this tool because the *p*-value was greater than 0.05 in one or both of the groups. The correlation factor between the RLT and the quiz results for the intervention and controls groups were Spearman 0.522, *p* < 0.00 and Spearman 0.536, *p* < 0.00, respectively. We measured radiology literacy for certain participant characteristics and found that radiology literacy increases as the educational level of the participants increases. There is a decline in awareness until the level of income reaches $20,000 to $29,999. At this point, the level of awareness begins to increase. Our results also shows that non-smokers scored 10.4 higher awareness than smokers. This founds to be consistent with prior studies reporting that general health literacy is associated with smoking [31,32]. People who had chronic diseases were more likely to have a lower radiology literacy than healthy people at a rate of 5.08. The chronic conditions included in the study were hyperlipidemia, high blood pressure, cancer, liver disease, diabetes, stroke, Alzheimer’s disease, and other chronic medical condition(s). Determining radiology knowledge and identifying where this knowledge is lacking is a key part in helping the health system address the knowledge deficit. For example, those who are middle-income, smokers, or have chronic diseases can be targeted to increase their radiology literacy levels. The tool can also be used for residents in rural areas to develop healthcare in those regions. The RLT tool devolved can have a wide range of practical applications, including the assessment of the impact of educational or technological interventions on patients’ radiology literacy. In addition to intervention pre- and post-assessments, RLT can be utilized to compare different solutions and intervention programs. For future studies, we also recommend applying the RLT for larger demographics to examine the tool’s reliability and generalizability.

## 6. Conclusions

This study provides a tool that can measure the extent of health awareness in the radiology field. The study is the first of its kind and the first step in developing tools to measure radiology awareness. The results of the study can be used to help specialists in the radiology field modify or create new reporting methods by providing specialists with more information about the public’s knowledge of radiology. The information from this study will allow patient needs to be better understood and better served by the medical community.

## Figures and Tables

**Figure 1 ijerph-19-11808-f001:**
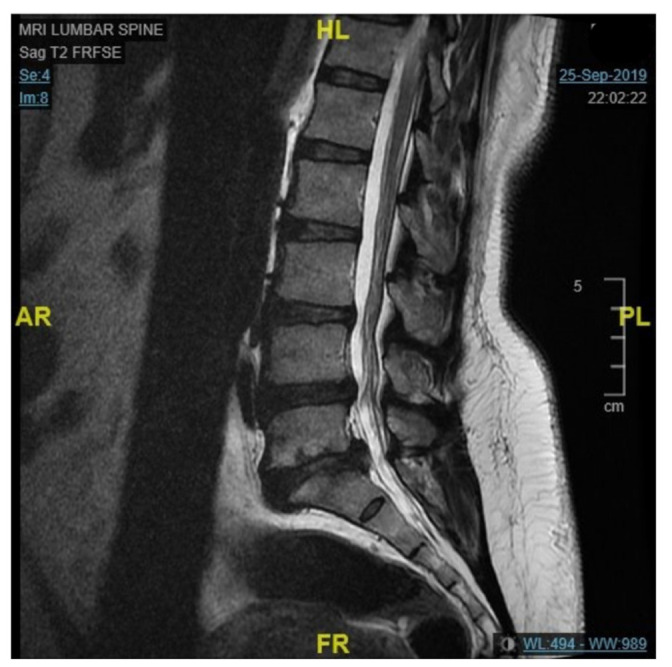
Original design of the MRI lumbar spine image.

**Figure 2 ijerph-19-11808-f002:**
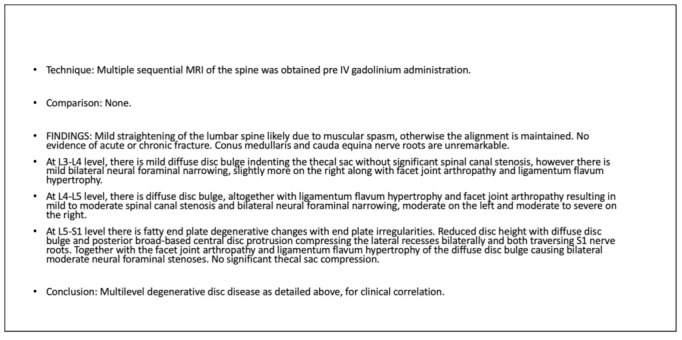
Original design of the MRI lumbar spine report.

**Figure 3 ijerph-19-11808-f003:**
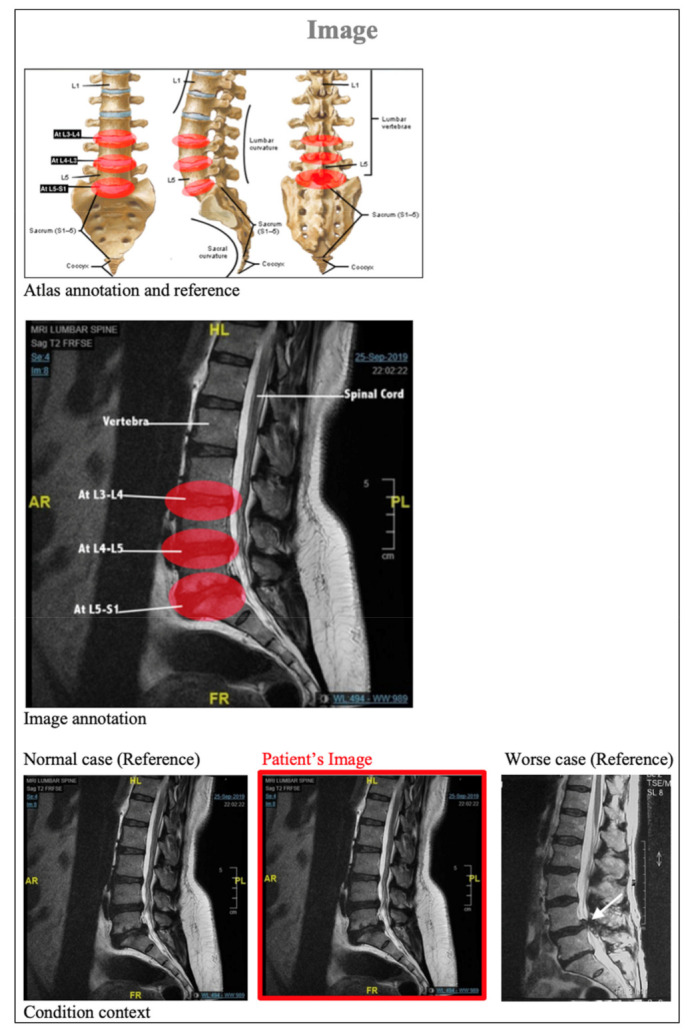
Revised design of the MRI lumbar spine image.

**Figure 4 ijerph-19-11808-f004:**
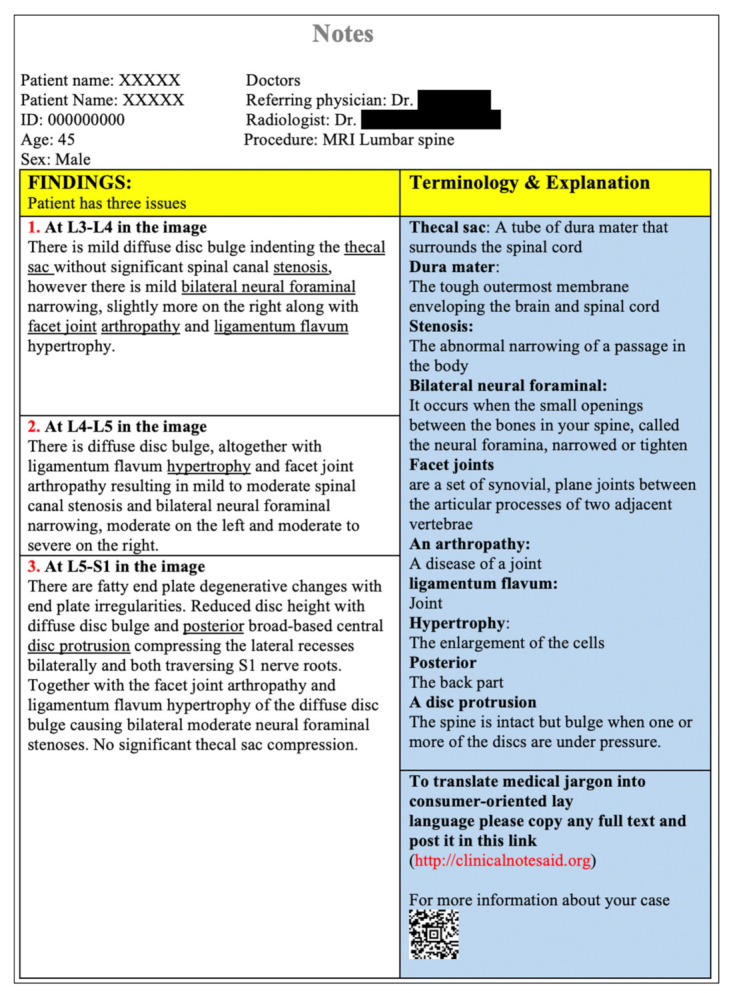
Revised design of the MRI lumbar spine report.

**Figure 5 ijerph-19-11808-f005:**
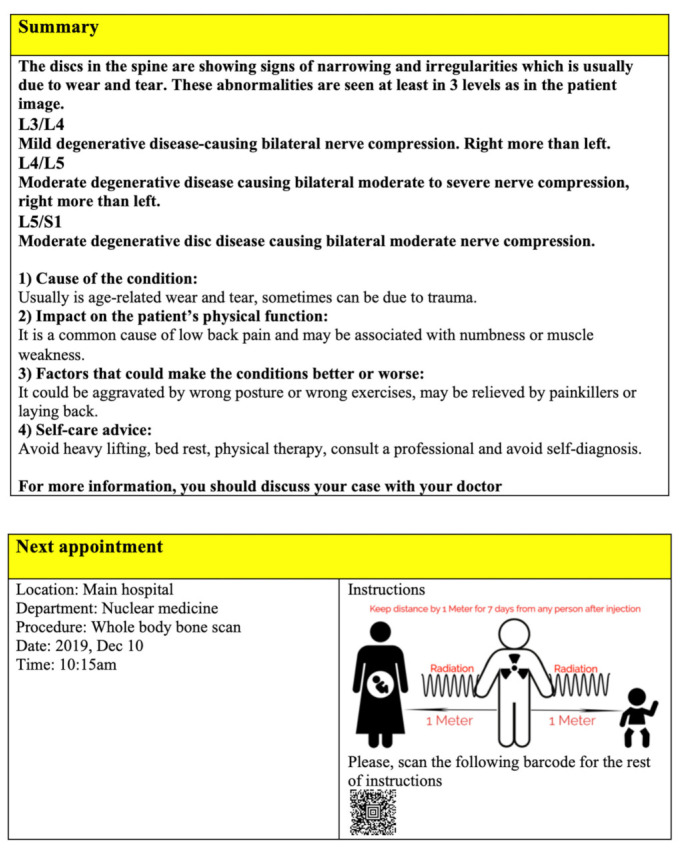
Revised design of the MRI lumbar spine report.

**Figure 6 ijerph-19-11808-f006:**
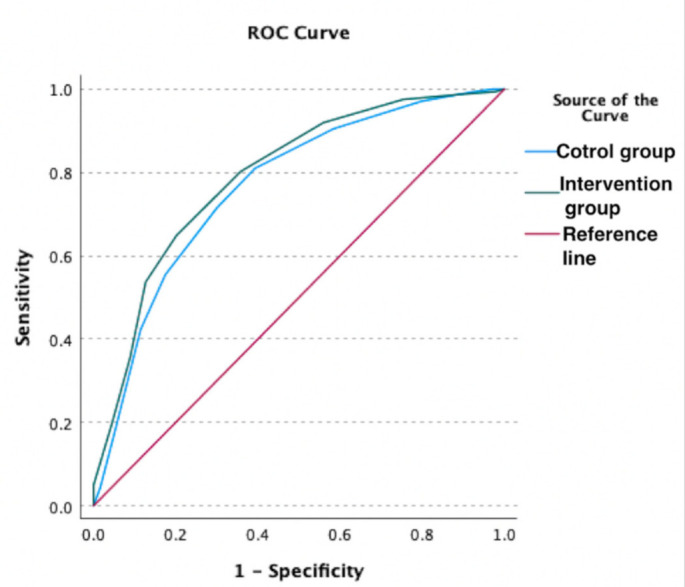
Receiver Operating Characteristics (ROCs) curve of patients’ understanding of the quiz and their radiology literacy outcomes for the control and intervention groups.

**Table 1 ijerph-19-11808-t001:** The correlation between each question in the RLT questions and RCQ results.

NO	Question	Control Group	Intervention Group	*p* Value
1	The radiological modality that uses a magnetic field to create images of the inside of your body is (CT, MRI, NM, don’t know)	(Spearman 0.358, *p* < 0.001)	(Spearman 0.249, *p* < 0.001)	0.553
2	The radiological modality that uses X-rays to create images of the inside of your body is (CT, MRI, NM, don’t know)	(Spearman 0.393, *p* < 0.001)	(Spearman 0.359, *p* < 0.001)	0.936
3	The radiological modality that uses small amounts of radioactive material to create images of the inside of your body is (CT, MRI, NM, don’t know)	(Spearman 0.312, *p* < 0.001)	(Spearman 0.335, *p* < 0.001)	0.091
4	All radiology modalities use radiation in the scans (yes, no, don’t know)	(Spearman 0.179, *p* 0.001)	(Spearman 0.212, *p* < 0.001)	0.513
5	CT uses radiation, which can cause cancer (yes, no, don’t know)	(Spearman 0.070, *p* 0.223)	(Spearman 0.015, *p* 0.838)	0.253
6	There is no limit to do many X-ray scans in per year (yes, no, don’t know)	(Spearman 0.286, *p* < 0.001)	(Spearman 0.321, *p* < 0.001)	0.872
7	The body can filter all the radiation from the body at the end of the imaging scan day (yes, no, don’t know)	(Spearman 0.114, *p* 0.042)	(Spearman 0.249, *p* < 0.001)	0.069
8	This is a kidney image (yes, no, don’t know) 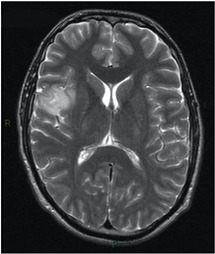	(Spearman 0.250, *p* < 0.001)	(Spearman 0.261, *p* < 0.001)	0.441
9	The case in the above image is (normal case, not normal case, don’t know)	(Spearman 0.109, *p* 0.062)	(Spearman 0.015, *p* 0.835)	0.298
10	It’s normal that radiology images appear in white and gray colors (yes, no, don’t know)	(Spearman 0.213, *p* < 0.001)	(Spearman 0.200, *p* 0.001)	1.00
11	All the imaging tests have the same preparation instructions (yes, no, don’t know)	(Spearman 0.251, *p* < 0.001)	(Spearman 0.343, *p* < 0.001)	0.104
12	The radiology scan that requires no metal on the body is (Ultrasound, MRI, don’t know)	(Spearman 0.276, *p* < 0.001)	(Spearman 0.183, *p* 0.002)	0.681
13	Did you know that radiology images can be provided in three views as in the above image? (yes, no, don’t know)	(Spearman 0.063, *p* 0.262)	(Spearman 0.018, *p* 0.796)	0.063

**Table 2 ijerph-19-11808-t002:** Demographic variables (interventional study).

Characteristics	Control Group		Intervention Group		*p* Value (Pearson Chi-Square)
**Age**					0.723
<20–29 years	39.4%	126	42.6%	256	
30–49 years	45.9%	147	43.2%	128	
50+ years	14.7%	47	14.2%	42	
**Gender**					0.014
Male	63.4%	203	53.2%	157	
Female	36.6%	117	46.8%	138	
**Level of education**					0.603
Some school and high school	8.8%	28	7.8%	23	
Some college	27.5%	88	31.1%	180	
College degree and above	63.7%	204	61.1%	385	
**English is the first language**					0.319
Yes	91.3%	292	88.9%	263	
No	8.8%	28	9.9%	33	
**Income**					0.733
$10,00	9.1%	29	10.8%	32	
$10,000–$19,999	12.5%	40	14.9%	44	
$20,000–$39,999	24.7%	79	19.9%	59	
$40,000–$59,999	22.2%	71	22%	65	
$60,000–$79,999	18.8%	60	18.6%	55	
≥$80,000	12.8%	41	13.9%	41	
**Chronic condition**					0.946
Yes	36.6%	117	36.8%	109	
No	63.45%	203	63.2%	187	
**Smoking**					0.832
Yes	39.7%	127	38.9%	115	
No	60.3%	193	61.1%	181	

**Table 3 ijerph-19-11808-t003:** RCQ questions to measure participant understanding.

No	Question	Choices	Score
1.	According to the report (Notes) the discs in the spine are	a. Fine b. Severely damaged c. Showing signs of narrowing and irregulates d. I do not know	1
2.	According to the report (image and notes), can you determine the location of the issues?	a. At L3 L4 b. At L4 L5 and L5 S1 c. At L3 L4, L4 L5, and L5 S1 d. I do not know	1
3.	According to the report (Image), can you determine the location of the L5 S1 in the following image 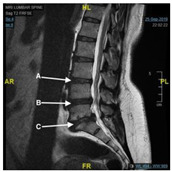	a. A b. B c. C d. I do not know	1
4.	According to the report (notes), the terminology word “Stenosis” means:	a. The abnormal narrowing of a passage in the body b. A disease of a joint c. Fever in the body d. I do not know	1
5.	According to the report (notes), the terminology word “Hypertrophy” means	a. Inflammatory condition of the liver b. Bone infection c. The enlargement of the cells d. I do not know	1
6.	Total score		5

**Table 4 ijerph-19-11808-t004:** The average RLT score stratified by different demographics with a total score of 100 (10 points per question).

Demographic Characteristics	Average Score	*p*-Value
**Level of education**		<0.001
Some school and high school	44.31	
Some college	46.28	
College degree and above	57.64	
**Income**		<0.001
<$10,00	51.97	
$10,000–$19,999	48.45	
$20,000–$39,999	47.46	
$40,000–$59,999	52.20	
$60,000–$79,999	59.82	
≥$80,000	61.10	
**Smoking**		<0.001
Yes	46.90	
No	57.30	
**Chronic diseases**		0.013
Yes	50	
No	55.08	

## Data Availability

The authors declare that they had full access to all of the data in this study and the authors take complete responsibility for the integrity of the data and the accuracy of the data analysis.

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
