# Peer review of "Proposed Questions to Assess the Extent of Knowledge in Understanding the Radiology Report Language"

_ijerph, 2022, doi:10.3390/ijerph191811808_

Round 1
Reviewer 1 Report
The topic of the paper is interesting and meets the aims and scope of the journal but it has some problems
However, I have some suggestions for the paper improvement, as follows:
Methods:
The methods are described in sufficient detail to understand the approach used but it is not clearly the aim and the research questions of the study. Also, there is not clearly mention regard to statistical tests were applied.
The questionnaire of this study has sufficient reliability and validity? It is not clear for the reader.
The results are well -written.
There is no mention into the text about the ethics of the study. You should mention the process of ethical approval of the study, if there is a consent for participants.
Also, you should write the limitations and the strenghts of this study.
Although the study is the first of its kind I would like to suggest presenting your significant results (in the discussion section) and interpret them appropriately.
Author Response
Reviewer 1
Dear editor
I would like to thank you and thank the reviewers for accepting the manuscript and the invitation to resubmit. We appreciate the reviewers precious time, thoughtful review and invaluable comments. We have carefully addressed all the comments which we believe have greatly help improve the manuscript. We edited the manuscript in responses to reviewers’ comments and provided the listed specific response to each comment below. Also, we paraphrase the abstract.
Reviewer Comment
Methods:
The methods are described in sufficient detail to understand the approach used but it is not clearly the aim and the research questions of the study. Also, there is not clearly mention regard to statistical tests were applied.
Response;
Thanks for your comment. We edited the manuscript to clarify the study aim and the statistical analysis.
Reviewer Comment
The questionnaire of this study has sufficient reliability and validity? It is not clear for the reader.
Response; Line 137
Thank you, in this study we conducted a face validation and we added a description of the validation process which we copied below;
“For validation, we adopted a Lawshe techniques for content and face validation. A group of radiologists and a radiology specialists were consulted to finalize the latest draft of the questions presented in our radiology knowledge assessment tool.”
For reliability we recommend future studies to adopt the tool developed and this will help examine both the reliability and generalizability of our finding. We added this recommendation to our writing at the end of the discussion section.
Reviewer Comment
The results are well -written.
There is no mention into the text about the ethics of the study. You should mention the process of ethical approval of the study, if there is a consent for participants.
Response;
We added more exculpations regarding the ethics. 295 – 279 AND
Conflicts of Interest: The authors declare no conflict of interest. The University of Wisconsin – Milwaukee Institutional Research Ethics Committee has confirmed that no ethical approval is required (#IRB 20.230). Line 673
Reviewer Comment
Also, you should write the limitations and the strenghts of this study.
Response:
Line 249-254, We added some points
Reviewer Comment
Although the study is the first of its kind I would like to suggest presenting your significant results (in the discussion section) and interpret them appropriately.
Response
In this revision, we did heavy enhancements in manuscript

Reviewer 2 Report
The paper studies how to measure the extent of radiology awareness. Two groups of patients are designed to evaluate their awareness of the radiology reports with different designs using correlation analysis between each question and the quiz grade. The paper is well-written and well-organized. There are some issues:
1. The introduction could be improved. More references about the subject could be added.
2. Questions are used to evaluate the understanding of patients for the reports. Would the design of the questions influence the results? More details of the design rules of the questions could be added.
3. What is the contribution of the paper? The new design of the radiology reports?
4. What is the specific meaning of the "tool"? The analysis method or the report design, or both?
Author Response
Reviewer 2
Dear editor
I would like to thank you and thank the reviewers for accepting the manuscript and the invitation to resubmit. We appreciate the reviewers precious time, thoughtful review and invaluable comments. We have carefully addressed all the comments which we believe have greatly help improve the manuscript. We edited the manuscript in responses to reviewers’ comments and provided the listed specific response to each comment below. Also, we paraphrase the abstract.
Reviewer Comment
- The introduction could be improved. More references about the subject could be added.
Response;
Thanks for your comments and we added some enhancements with more citations in the introduction.
Reviewer Comment
- Questions are used to evaluate the understanding of patients for the reports. Would the design of the questions influence the results? More details of the design rules of the questions could be added.
Response; Line 132-142
Thanks for this suggestion, we added the survey design rules adopted in our study to the method section. Copy of the relevant part added is copied below;
“The questions worded carefully to avoid biased, leading questions, assumptions, or loaded questions which might influence the results. The questions were also reviewed multiple times and tested following the design principles published by Iarossi, G. (2006).”
Iarossi, G. (2006). The power of survey design: A user's guide for managing surveys, interpreting results, and influencing respondents. World Bank Publications.
Reviewer Comment
- What is the contribution of the paper? The new design of the radiology reports?
Response;
In line 657 we explain the contribution of the RLT and the effectiveness of enhancing the public health services
Reviewer Comment
- What is the specific meaning of the "tool"? The analysis method or the report design, or both?
Response;
RLT Radiology Literacy Tool line 42
Also, we added more explanation in line 131
“To develop the RLT, a set of questions to measure the extent of knowledge of medical radiology in patients is presented in (Table 1).”

Reviewer 3 Report
The manuscript entitled "Proposed Questions to Assess the Extent of Knowledge in Understanding the Language of the X-ray Report" reports the assessment of a tool for measuring patient literacy. The authors already presented the tool in a previous article. The argument treated can be of interest, but the manuscript highlighted several issues in the key sections. In particular, the description of the methods is not clear, and the statistical analysis is very poor. My recommendation is to review these sections and then resubmit the manuscript. A non-exhaustive list of details is provided below.
Line 76. Tables should be numbered consecutively with the text.
Line 83. The Pearson correlation is inappropriate for discrete variables with few values (in particular, they are prevalently nominal). In this case, a non-parametric test is required. Moreover, in order to verify differences between groups, for each variable reported in Table 3, the two groups should be statistically compared.
The authors declared “The participants were divided by a computer into two groups” (line 84). Subjects randomly assigned to the two groups should have a very similar size; instead, the size was 320 and 296, respectively for the control and intervention groups.
It is not understandable as was obtained the accuracy of the tool (line 88). For example, in the ref. [15] indicated from the authors is clearly reported that “a score of ten was assigned for each correct response”. Moreover, in this article, the score was plotted versus the percentage of the respondents (fig. 1, ref. [15]). Again, the percentage of survey respondents with correct responses to questions, also comparing the groups, was reported (Table 2, ref. [15]). At the last, the average hypertension knowledge score comparing the two groups was also given (Table 3, ref. [15]). Regarding the manuscript, in Table 4 caption is reported (line 154): “This method used in a previous study [1].” in this last reference, the method did not describe.
Line 105. For the sample size calculation, the alpha usually is 0.05 (probably a typo).
Line 109. In Table 2, as well as in Table 3, the authors did not verify the significance between groups. Moreover, the Smoking variable seems not appropriate, because in the manuscript were no reported images or terms related to tobacco and associated diseases. Finally, the “Chronic condition” variable should be defined.
Line 138. P-values should be reported with at least one significant digit, or as an upper limit (e.g. P <.001).
Line 138. It is not clear as the ROC curve was calculated.
Author Response
Reviewer 3
Dear editor
I would like to thank you and thank the reviewers for accepting the manuscript and the invitation to resubmit. We appreciate the reviewers precious time, thoughtful review and invaluable comments. We have carefully addressed all the comments which we believe have greatly help improve the manuscript. We edited the manuscript in responses to reviewers’ comments and provided the listed specific response to each comment below. Also, we paraphrase the abstract.
Reviewer Comment;
Line 109. In Table 2, as well as in Table 3, the authors did not verify the significance between groups. Moreover, the Smoking variable seems not appropriate, because in the manuscript were no reported images or terms related to tobacco and associated diseases. Finally, the “Chronic condition” variable should be defined.
Response;
Thank you for your comment. In line 525 Table 2, we added the a new Column for the significant. Also, Table 4 (Table 3 before) we added the same point.
The smoking status was included in the demographics to explore as one of the demographics which might be related to patients’ literacy similar to levels of education, and income. It was not intended to be relevant to specific condition in the radiology report.
Some studies suggested that smoking status can be relevant to patients literacy and because of that we included it to check if the same thing will apply to radiology literacy
Thank you for your comment,
We added the chronic condition variables in line 643
Comment;
Line 76. Tables should be numbered consecutively with the text.
Response;
Good suggestion, we modified the manuscript based on the reviewer comment and organized all tables consecutively with the text.
Reviewer Comment
Line 83. The Pearson correlation is inappropriate for discrete variables with few values (in particular, they are prevalently nominal). In this case, a non-parametric test is required. Moreover, in order to verify differences between groups, for each variable reported in Table 3, the two groups should be statistically compared.
Response;
Thanks for this suggestion, we repeated the analysis and performed a non-parametric test (spearman) instead of Pearson and updated all the related results throughout the manuscripts text and tables.
With regard to the second part of the reviewer comment. We also added the p-value to statistically compare the different between groups in Table 3. please note that table 3 is now labeled as Table 1 Line 520
Reviewer Comment
The authors declared “The participants were divided by a computer into two groups” (line 84). Subjects randomly assigned to the two groups should have a very similar size; instead, the size was 320 and 296, respectively for the control and intervention groups.
Response;
Great comment, after the survey collection process, we found that some of the respondents did not complete all the questions. Those with missing data were excluded from the analysis and thus we had uneven groups.
Reviewer Comment
It is not understandable as was obtained the accuracy of the tool (line 88). For example, in the ref. [15] indicated from the authors is clearly reported that “a score of ten was assigned for each correct response”. Moreover, in this article, the score was plotted versus the percentage of the respondents (fig. 1, ref. [15]). Again, the percentage of survey respondents with correct responses to questions, also comparing the groups, was reported (Table 2, ref. [15]). At the last, the average hypertension knowledge score comparing the two groups was also given (Table 3, ref. [15]). Regarding the manuscript, in Table 4 caption is reported (line 154): “This method used in a previous study [1].” in this last reference, the method did not describe.
Response;
Thanks for your comment, we belief the word “accuracy” in our previous writing may not reflect the intended meaning, therefor we rewrote the previous paragraph to better describe that part of the analysis which we copied below;
“To identify the RLT questions indicating patients’ literacy, we assessed the correlation between each of the 13 questions in the RLT (Table 3) with the total score of RCQ questions (Table 1) using Spearman correlation coefficients. Three questions were ex-cluded from the RLT question list (questions 5, 9, and 13) because the P-value for the questions was greater than 0.05 (Table 1). Also, we used Receiver Operating Charac-teristic ROC to compute sensitivity and specificity of the RLT. There were two variables we used, the total RCQ score and the total RLT score. The RCQ results were divided into two groups based on the total score: 0–2 and 3-5 (Figure 6). To calculate the total RLT score, we assigned a value of 10 for each question, giving the respondent a possible total of 100 points. Similar calculation was used in other studies to measure literacy in various health fields [12-15]. The statistical analyses were performed using SPSS software.”
Reviewer Comment
Line 105. For the sample size calculation, the alpha usually is 0.05 (probably a typo).
Response
Thanks for your comment and we corrected the typo (Line 288)
Reviewer Comment
Line 138. P-values should be reported with at least one significant digit, or as an upper limit (e.g. P <.001).
Response
Thanks for your comments, we corrected the p value in line 577
Reviewer Comment
Line 138. It is not clear as the ROC curve was calculated.
Response
Thanks for your comments, we add more explanations on ROC in line 308
Also, we added more analysis on ROC.

Round 2
Reviewer 3 Report
The manuscript is improved, but the following topics should be resolved.
---
Authors’ response: Some studies suggested that smoking status can be relevant to patients literacy and because of that we included it to check if the same thing will apply to radiology literacy.
The authors should report this affirmation in the manuscript, and to support it should insert the studies in the references.
---
Authors’ response: Thanks for this suggestion, we repeated the analysis and performed a non-parametric test (spearman) instead of Pearson and updated all the related results throughout the manuscripts text and tables.
In general, the variables are nominal (see previous revision); for example, “yes, no, don’t know”. This means that they have no natural order, and parametric or not parametric correlation is inappropriate. “In this case, a non-parametric test is required”. The authors should perform a no parametric test; for example, chi-squared between groups (see your reference: Li, X., et al., Health literacy in rural areas of China… However, in table 1 the comparison between groups is missing, and P-values are again reported as “.00”.
---
Authors’ response: Great comment, after the survey collection process, we found that some of the respondents did not complete all the questions. Those with missing data were excluded from the analysis and thus we had uneven groups.
This procedure is not right. In order to obtain a correct random assignment to the two groups, the authors should carry out the subjects' assignation after the responses checking. In this way, the groups would have a very similar sample size, which is a more realistic condition for the statistical outcomes.
Author Response
Reviewer 3
Dear editor
I would like to thank you again and thank the reviewer for accepting the manuscript and the invitation to resubmit. We appreciate the reviewer precious time, thoughtful review and invaluable comments. We have carefully addressed all the comments which we believe have greatly help improve the manuscript. We edited the manuscript in responses to reviewer’ comments and provided the listed specific response to each comment below. Al
The manuscript is improved, but the following topics should be resolved.
---
Comment 1
Authors’ response: Some studies suggested that smoking status can be relevant to patients literacy and because of that we included it to check if the same thing will apply to radiology literacy.
The authors should report this affirmation in the manuscript, and to support it should insert the studies in the references.
Response:
Thank you, we edited the manuscript to better describe this point and add the relevant references which can be found in the discussion section. (Line 651)
Comment 2
Authors’ response: Thanks for this suggestion, we repeated the analysis and performed a non-parametric test (spearman) instead of Pearson and updated all the related results throughout the manuscripts text and tables.
In general, the variables are nominal (see previous revision); for example, “yes, no, don’t know”. This means that they have no natural order, and parametric or not parametric correlation is inappropriate. “In this case, a non-parametric test is required”. The authors should perform a no parametric test; for example, chi-squared between groups (see your reference: Li, X., et al., Health literacy in rural areas of China… However, in table 1 the comparison between groups is missing, and P-values are again reported as “.00”.
Response:
Thanks for your detailed clarification,
In response to your suggestion, we added a non-parametric chi-square test to compare groups Table 1 Line 532
We also would like to note the following clarification related to the non-parametric Spearman reported in the same table. It was done to examine the correlation between the following variables: every question of the RLT (dichotomous; correct or incorrect answer) and the total score of RCQ (continuous from 1 to5).
Also, we changed the P-values to the write way.
Comment 3
Authors’ prior response: Great comment, after the survey collection process, we found that some of the respondents did not complete all the questions. Those with missing data were excluded from the analysis and thus we had uneven groups.
This procedure is not right. In order to obtain a correct random assignment to the two groups, the authors should carry out the subjects' assignation after the responses checking. In this way, the groups would have a very similar sample size, which is a more realistic condition for the statistical outcomes.
Response:
Participants were electronically randomized immediately after completing the common section and we were unable to carry out the assignments after checking the results. While this was not ideal, we examined the data by comparing the groups characteristics (Table 2) to check for any systematic differences and if randomization is truly preserved.
In the manuscript we add more details (Line 301-305)
McHugh, M. L. (2013). The chi-square test of independence. Biochemia medica, 23(2), 143-149.
Also, beside that this paper is part of project we published
Alarifi, M., Patrick, T., Jabour, A., Wu, M., & Luo, J. (2021). Designing a consumer-friendly radiology report using a patient-centered approach. Journal of Digital Imaging, 34(3), 705-716.